# Insightful Improvement in the Design of Potent Uropathogenic *E. coli* FimH Antagonists

**DOI:** 10.3390/pharmaceutics15020527

**Published:** 2023-02-04

**Authors:** Leila Mousavifar, Meysam Sarshar, Clarisse Bridot, Daniela Scribano, Cecilia Ambrosi, Anna Teresa Palamara, Gérard Vergoten, Benoît Roubinet, Ludovic Landemarre, Julie Bouckaert, René Roy

**Affiliations:** 1Glycosciences and Nanomaterial Laboratory, Université du Québec à Montréal, Succ. Centre-Ville, P.O. Box 8888, Montréal, QC H3C 3P8, Canada; 2Research Laboratories, Bambino Gesù Children’s Hospital, Istituto di Ricovero e Cura a Carattere Scientifico, 00146 Rome, Italy; 3Unité de Glycobiologie Structurale et Fonctionnelle (UGSF), UMR8576 du CNRS, Univ. Lille, 59000 Lille, France; 4Department of Public Health and Infectious Diseases, Sapienza University of Rome, 00185 Rome, Italy; 5Department of Human Sciences and Promotion of the Quality of Life, San Raffaele Roma Open University, 00166 Rome, Italy; 6Istituto di Ricovero e Cura a Carattere Scientifico, San Raffaele Roma, 00166 Rome, Italy; 7Department of Infectious Diseases, National Institute of Health, 00161 Rome, Italy; 8Institut de Chimie Pharmaceutique Albert Lespagnol (ICPAL), Faculté de Pharmacie, University of Lille, 3 Rue du Professeur Laguesse, BP-83, 59006 Lille, France; 9GLYcoDiag, 2 Rue du Cristal, 45100 Orléans, France

**Keywords:** uropathogenic *Escherichia coli* (UPEC), FimH, antagonists, mannosides, glycocomimetics, crystallography, adhesion inhibition, bladder cells, molecular dynamic simulations

## Abstract

Selective antiadhesion antagonists of Uropathogenic *Escherichia coli* (UPEC) type-1 Fimbrial adhesin (FimH) are attractive alternatives for antibiotic therapies and prophylaxes against acute or recurrent urinary tract infections (UTIs) caused by UPECs. A rational small library of FimH antagonists based on previously described *C*-linked allyl α-D-mannopyranoside was synthesized using Heck cross-coupling reaction using a series of iodoaryl derivatives. This work reports two new members of FimH antagonist amongst the above family with sub nanomolar affinity. The resulting hydrophobic aglycones, including constrained alkene and aryl groups, were designed to provide additional favorable binding interactions with the so-called FimH “tyrosine gate”. The newly synthesized *C*-linked glycomimetic antagonists, having a hydrolytically stable anomeric linkage, exhibited improved binding when compared to previously published analogs, as demonstrated by affinity measurement through interactions by FimH lectin. The crystal structure of FimH co-crystallized with one of the nanomolar antagonists revealed the binding mode of this inhibitor into the active site of the tyrosine gate. In addition, selected mannopyranoside constructs neither affected bacterial growth or cell viability nor interfered with antibiotic activity. *C*-linked mannoside antagonists were effective in decreasing bacterial adhesion to human bladder epithelial cells (HTB-9). Therefore, these molecules constituted additional therapeutic candidates’ worth further development in the search for potent anti-adhesive drugs against infections caused by UPEC.

## 1. Introduction

One of the major pathogens responsible for urinary tract infections (UTIs) is uropathogenic *Escherichia coli* (*E. coli*) (UPEC), leading to a major burden in public health [1]. As part of the normal microbiota, *E. coli* exhibits diverse species with a wide spectrum of phenotypes that reside in the large intestine of humans and many animals [2]. Several virulence factors are responsible for the establishment of current infections [2,3]. While *E. coli* mainly live harmlessly in the gut as their primary niche by establishment of a symbiotic relationship, some strains act as pathogens and cause a variety of enteric and extra-enteric diseases [4,5,6]. Among these, the adhesion of UPECs to the host uroepithelial tissues is due to the *E. coli* type 1 pili, called FimH, which mediate their attachment in a mannose-dependent interaction [4,5]. FimH adhesin of type 1 fimbriae promotes intestinal colonization by binding to colonic crypts of epithelial cells, facilitating *E*. *coli* adhesion and invasion to the urinary epithelium, a key player in UPEC pathogenicity [6]. Preventing bacterial FimH adhesion to highly mannosylated glycoprotein uroplakin Ia from bladder cells represents an appealing strategy to control UTIs [4]. Since bacterial adhesion to host cells is a critical and initial step in most infectious diseases, anti-adhesion therapy can effectively compete against antibiotic therapy and is seriously considered for the treatment of acute uncomplicated lower UTIs (AUC) [7,8,9].

Importantly, the mannose-specific adhesin FimH has an essential hydrophobic binding pocket of non-polar amino acids that surrounds the aglycon portion of α-D-mannopyranoside glycomimetics. These findings have been fully exploited in the design of potent FimH anti-adhesins [10,11,12,13]. Glycomimetics are simplified representatives of more complex glycoconjugate structures found in nature. They have been designed to improve carbohydrate-protein binding interactions and to provide better pharmacokinetic and pharmacodynamic properties [14,15]. 

As alternative therapeutic strategies against UTIs, monomeric mannoside antagonists [15,16,17,18] and clusters [19,20,21,22,23,24,25] with varied hetero-anomeric linkages *O*-[26,27,28,29], *S*-[30,31,32], or *N*-[33,34] have been developed as appealing candidates. Using a plethora of available crystallographic data, detailed structure-activity relationships (SARs) have been established to provide further insight into the binding mode of *E. coli* FimH. This approach has greatly helped in improving the conception of potent antagonists [4,35,36,37,38,39]. Amid the various α-D-mannopyranoside inhibitors described thus far, *C*-linked glycomimetics residues harboring hydrophobic aglycons are still considered worthy candidates as potent FimH inhibitors. Several of these synthetic candidates showed improved binding affinities and increased hydrolytic stability [30,40,41].

This work describes the synthesis, relative binding affinities, crystallographic data, and anti-adhesive properties of a small library of *C*-linked mannopyranoside inhibitors. The data provide further pivotal insights into the potential of these glycomimetics against UPEC strains and give rise to a solid basis toward the development of new and effective FimH antagonists.

## 2. Results and Discussion

### 2.1. Synthesis and Structural Characterization

In previous studies, several aryl *C*-mannopyranosides carrying alkenyl aglycons were shown as promising candidates with excellent binding affinities against *E. coli* FimH [40,41]. However, their poor water solubility has restricted their further development as drug candidates. To address this issue, we propose herein the use of heteroaryl-substituted mannopyranosides based upon numerous available crystallographic data.

The first step toward this next generation of antagonists was the improved preparation of *C*-allyl α-D-mannopyranoside **1** as a key starting material. Compound **1** was obtained by per-*O*-benzoylation of commercially available methyl α-D-mannopyranoside (MeMan) followed by a Hosomi–Sakurai reaction to afford the pure α-anomer **1** [40]. Key precursor **1** was next treated under palladium-catalyzed Heck conditions in the presence of aryl iodides (DMF, TBAB, NaHCO_3_, Pd(OAc)_2_, 90 °C, o.n.) to afford a family of (*E*)-linked aryl derivatives (**2**–**10**) in good yields (Figure 1).

As depicted in Appendix A, ^1^H-NMR spectra (600 MHz) of both compounds **2** and **11,** together with detailed 2D NMR spectrum of compound **2**, unambiguously confirmed that all the major compounds were the *trans* stereoisomers. Reliable evidence of this claim was the signal of H-3’ of the alkene that appeared as a doublet at δ 6.67 ppm with a *J*_2’,3’_ = 15.8 Hz and 16 Hz for both compounds **2** and **11**, respectively. Moreover, the coupling constants for the H4 signal at δ 6.00 ppm (dd, 1H, *J*_3,4_ = *J*_4,5_ = 8.5 Hz, H-4) unambiguously indicated a *trans*-diaxial relationship between H3-H4 and H4-H5, thus confirming this series of compounds to be in the proper ^4^*C*_1_ chair conformation.

Because *C*-linked mannopyranosides do not exhibit an anomeric effect in comparison to their *O*-linked analogs, there was a risk that the analogs had undergone conformational changes from the desired ^4^*C*_1_ chair to skew boats (^0^S_2_ or _0_S^2^) [4]. The X-ray crystal structure of perbenzoylated compound **2** showed it to be in the desired chair conformation as previously observed with several related analogs (Appendix A). Since the conformation of **2** in the solid-state could be different than that in solution, an in-depth NOESY analysis was also performed to further confirm its conformation (Appendix A). The nOe between the geminal H1’a, H1’a’ with the axial H3 and H5 unmistakably implied the ^4^*C*_1_ conformation. Analysis of the ^1^H-NMR spectrum of *C*-linked mannosides (**11**–**18**, and **20**) also showed values of ca. 1.63 Hz for their ^3^*J*_1,2_ coupling constants of their vicinal equatorial/axial arrangements (Appendix A).

Finally, benzoyl ester deprotection of *C*-linked mannopyranosides (**2**–**9**) was performed using ammonia in methanol (1M, r.t., o.n.) to provide unprotected derivatives (**11**–**18** and **20**), essentially quantitatively.

### 2.2. Synthesis of a Key Ortho-Substituted Biphenyl Derivative

In our previous work, an *ortho*-substituted *C*-linked biphenyl mannopyranoside candidate was considered as a promising lead with a *K*_D_ of 6.9 nM [41]. Unfortunately, this compound had poor water solubility. To further exploit this promising series of analogs, we envisaged the insertion of heteroaryl moieties at the *ortho* position in order to improve their water-solubility. Hence, compound **10** was synthesized under the above Heck reaction conditions using 1-bromo-2-iodobenzene (DMF, TBAB, NaHCO_3_, Pd(OAc)_2_, 90 °C, o.n.). This intermediate was then treated under Suzuki–Miyaura [42,43] coupling conditions using 4-pyridylboronic acid (cesium carbonate, [1,1’-bis(diphenylphosphino)ferrocene] dichloropalladium(II), 80 °C, dioxane/water, 5:1, o.n.) to give compound **19** in 55% yield. Compound **19** was deprotected using a solution of ammonia (1M) in methanol (r.t., o.n.) to afford compound **20** in 80% yield (Figure 2).

### 2.3. Affinity Evaluation of Mannosides through FimH LEctPROFILE Kit

To measure the relative binding affinity of the newly synthesized compounds **11**, **20**, and **18** against FimH, the usual referenced standards were used for comparison with previously published data: heptyl α-D-mannopyranoside (HepMan, **21**), *para*-nitrophenyl α-D-mannopyranoside (PNPMan, **22**), and methyl α-D-mannopyranoside (MeMan, **23**) [4]. The relative inhibitory potential of the above compounds was determined against a biotinylated Man-BSA conjugate using a FimH lectin domain (amino acids 1-158) LEctPROFILE kit (provided by GLYcoDiag) (Figure 1, Table 1). In this assay, IC_50s_ of 19.4 nM, 74.1 nM, and 2810.7 nM, were respectively obtained for **21**, **22**, and **23**. As Figure 1 shows, the best inhibition was obtained from the inhibitor **20**, followed by **11**, and finally the *C*-linked mannopyranoside **18** with IC_50s_ of 0.82 nM, 3.17 nM, and 30.28 nM, respectively. Undoubtably, the *ortho*-substituted pyridyl derivative **20** is the most efficient monomeric antagonist identified thus far. Interestingly, and as anticipated, compound **20** showed a calculated LogP (cLogP) improved (1.66) in comparison to its fully aromatic counterparts **18** (3.16).

### 2.4. X-ray and Molecular Dynamic Simulations

Unfortunately, no crystalline structure could be obtained from our best inhibitor **20**. However, co-crystallized FimH with the second-best inhibitor **11** was obtained at a resolution of 3 Å (Appendix A, PDB entry code 8BVD). The structure was solved using the PHENIX [44] software to visualize the detailed binding interactions which took place between the FimH binding domain and the inhibitor.

Delightfully, the co-crystal structure of compound **11** and *E. coli* C43 (DE3) FimH [45,46] also revealed the expected ^4^*C*_1_ chair conformation of the ligand **11** bound into the active site of the tyrosine gate. In the mannose-binding site of FimH, we observed Tyr48 in parallel, while the Tyr137 residue was in T-aromatic stacking with the quinoline group of ligand **11**. Four FimH lectin domains were observed in the asymmetric unit of the crystal (Appendix A) and the mannose-binding site of each FimH lectin domain was in proximity with a symmetry-related neighbor in the crystal packing (Figure 2) [34]. The two binding sites of FimH molecules were held together by the two quinoline substituents crossed over in a parallel stacking (Figure 2), while their mannosides projected into the binding pocket of FimH. The distance between the axial O2 hydroxyl group of the non-reducing end mannoside was 12 A° which exactly matched the distance previously observed in divalently bound trimannoside to FimH (crystal structures with PDB entry codes 6GTV and 6GTW). Therefore, the binding of FimH to compound **11** appeared to simulate the bivalent binding of the natural oligomannosidic *N*-glycan structures [47]. Inhibitor **11** interacted via different amino acids that included Ile 13, Phe 1, Asp 47, Gln 133, Asp 54, Asn 135, Asp 140, Phe 142 in the open conformation of FimH (Figure 3).

#### Further Insights into the Design and Binding of Compound **20**

Given that the best compound **20** failed to crystallize with the FimH lectin domain, we tried to explain its improved potency using molecular dynamics (MD) simulations. The best low energy score was obtained from its docking within the FimH half-open tyrosine gate (−90.70 kcal/mol) (PDB 4AUY) (Figure 4). For comparison, the lowest FimH closed and open energy conformations were at −60.34 and −89.76 kcal/mol, respectively [48,49,50,51]. 

According to previous work [41], the ortho-biphenyl substituted C-linked mannopyranoside with K_D_ = 6.9 ± 5.7 nM had a higher affinity over the para-substituted analog (compound **18**, Table 1) with K_D_ = 17 ± 3.5 [41]. Previously published MD simulation with an ortho-substituted biphenyl derivative could nicely explain the origin of the better affinity observed with **20**. Indeed, the MD simulation allowed us to postulate that the higher affinity might originate from a π-stacking between the first phenyl ring of **20** with Tyr48, while the second ortho pyridyl moiety can interact with the hydroxyl group of Tyr137 through a hydrogen bond (2.70 A°). Thus, the MD simulation was instrumental in the replacement of the second phenyl group with a heterocyclic pyridyl moiety to improve the solubility as well as the affinity of ligand **20**.

### 2.5. Mannosides Do Not Affect Bacterial Growth, Cell Viability, and Antibiotic Activities

To test whether the above mannoside derivatives could exert bactericidal and/or cytotoxic activities, bacterial growth and cell viability assays were undertaken in the presence of each mannoside at different concentrations. Natural D-mannose (D-Man) was included as control. As shown in Figure 5, bacteria grown in the presence of mannoside analogs showed similar growth rates compared to bacteria grown in LB (Luria–Bertani) medium that represents the positive control for bacterial replication. Moreover, no cytotoxic effect was recorded for urinary bladder cell line 5637 (HTB-9 cell) monolayers [48,52], as measured by the MTT test (Figure 6). Since D-mannose can serve as nutrient replacement for *E. coli* when there is a shortage of D-glucose, we tested whether these new derivatives could be used as carbon sources as well [5]. To further this aim, bacteria were cultured in LB medium for 12 h and then diluted to 1.5 × 10^7^ colony forming unit/mL (CFU/mL) in PBS buffer. Bacteria were incubated for an additional 24 h in the absence and in the presence of different mannosides and the number of viable bacteria was assessed by CFU/mL counting. Results show no statistically significant differences in the number of mannoside-treated bacteria compared to non-treated control (Figure 7). Conversely, the number of bacteria increased significantly in the presence of D-mannose, at both concentrations tested (500 μM and 83 mM) (Figure 7) [5]. The results indicated that the synthetic mannoside antagonists did not enter into the metabolic cycle of the bacterial cells to support their growth even in the absence of any other carbon sources. Differently, D-mannose as a natural sugar molecule is metabolized by bacteria, thereby maintaining bacterial replication [5]. Since carbohydrates can influence the activity of conventional antibiotics, the activity of different classes of antibiotics such as ampicillin (AMP 30 μg/mL), streptomycin (SM 50 µg/mL), and gentamycin (GM 50 µg/mL) in the presence of the synthetic mannoside analogs was evaluated. A broth-dilution test showed no differences in bacterial susceptibility, irrespectively in the presence of *C*-mannoside antagonists (Figure 8).

Altogether these results demonstrate the lack of toxicity of the synthetic mannoside derivatives toward both bacteria and eukaryotic cells. Moreover, these molecules are not used for bacterial metabolism and, unlike natural D-mannose, do not favor bacterial replication.

### 2.6. C-Mannoside Antagonists Are Effective in Decreasing Bacterial Adhesion to Human Bladder Epithelial Cells

To evaluate the efficacy of the synthetic mannoside antagonists to inhibit the ability of CFT073 strain to adhere to epithelial cells, an in vitro adhesion assay was performed. For this purpose, equal amounts of strain CFT073 strain were inoculated in PBS supplemented with different concentrations of each mannoside at final concentrations of 100, 500 μM, and 1 mM and incubated for 3h under static conditions. Bacterial inoculation was used to infect cell line 5637 (HTB-9 cell) monolayers at a multiplicity of infection (MOI) of 10. At 2.5 h post-infection, the number of cell-associated bacteria was calculated by CFU/mL counting (Figure 9). No significant inhibition of bacterial adhesion to bladder cells was obtained using mannoside inhibitors at 100 µM. Conversely, a significant reduction (more than 1 log) in the number of adherent bacteria was observed by increasing the concentration of the mannosides to 500 μM in comparison to non-treated bacteria (Figure 10). The same extent of reduction in bacterial adhesion to bladder cells was obtained by increasing mannosides concentration to 1 mM, thereby showing a dose-dependent effect. Interestingly, among the mannosides studied, the quinoline analog **11**, having an IC_50_ of 3.17 nM, was the most efficient at reducing FimH-mediated bacterial adherence (Figure 9 and Figure 10). On the other hand, compound **23** (MeMan) was the less efficient; however, it achieved the same extent of bacterial adhesion inhibition of natural D-mannose but at a 164-fold lower concentration. These results nicely confirmed previous observations from the control reference compounds **21**, **23**, and D-mannose [46]. To appraise qualitatively the reduction of bacterial adhesion, parallel infected cells were fixed, and Giemsa stained. As shown in Figure 10, no macroscopic differences in the shape, integrity, adhesiveness, cytoplasmic vacuolization, proliferation, or cytotoxic effects were observed in HTB-9 cell monolayers incubated with the synthesized mannosides, in line with the results of the MTT assay [5]. Overall, these results revealed that these synthetic inhibitors finely mimic the interaction between FimH and its natural receptor, thereby significantly decreasing the adhesion of strain CFT073 to bladder cells.

## 3. Materials and Methods

### 3.1. General Information

Nuclear magnetic resonance (NMR) spectra were recorded on Varian Geminin 300 MHz or Innova 600 MHz spectrometer. Chemical shifts are reported in parts per million (ppm) (δ) relative to CDCl_3_ (δ 7.27 and 77.23 ppm for ^1^H- and ^13^C-NMR, respectively) or relative to the signal of CD_3_OD (δ. 3.31 and 4.8 ppm for ^1^H- and 49.7 ppm for ^13^C-NMR, respectively). Where necessary, DEPT, APT, and two-dimensional 1H-1H COSY and HSQC experiments were performed for complete signal assignments. Coupling constants (J) are reported in Hertz (Hz). Signal multiplicities are used as singlet (s), doublet (d), doublet of doublet (dd), triplet (t), multiplet (m). Accurate mass measurements were performed on a LC-MSD-Tof instrument from Agilent technologies in positive electrospray with protonated ions [M + H]^+^; sodium adducts [M + Na]^+^ and [M + NH_4_]^+^ were used for empirical formula confirmation. Flash chromatography was performed using Merck silica gel 60 (40–63 µm). TLC was performed on Kiesel gel 60 F254 plates from Merck. Detection was carried out under UV light or by spraying with 20% ethanolic sulfuric acid or molybdate solution followed by heating. Optical rotations were measured with a JASCO P-1010 polarimeter. Melting points were measured on a Fisher Jones apparatus and are uncorrected. Purification of some compounds were done by semi-preparative HPLC (Agilent, Santa Clara, CA, USA). All eluents contained 0.1% formic acid and flow rate was set to 5 mL/min. Solvents were dried by distillation from drying agents as follows: DMF (Barium oxide), CH_2_Cl_2_ (P_2_O_5_), Et_3_N and pyridine (CaH_2_), MeOH was stored over 4A molecular sieves. Some aryl iodides were synthesized according to literature procedures [53,54,55]: 2-iodophenyl acetate, 4-iodophenyl acetate, 4-iodobenzyl acetate and methyl 4-iodobenzoate. Compound **9** and **18** were prepared as previously published [30]. 

### 3.2. Synthetic Methods and Analytic Data of Compounds

#### 3.2.1. General procedure for Heck Coupling of Protected C-Mannopyranosides **2–10**

To a solution of mannopyranosides **1** [40] in degassed anhydrous DMF, were added the various iodoaryl derivatives (2 equiv.), 10% palladium(II) acetate, tetrabutylammonium bromide (1 equiv.), and sodium bicarbonate (3 equiv.). The reaction mixture was heated at 85 °C under N_2_. The course of the reactions was followed by TLC. The solution was evaporated under reduced pressure and the residue was purified by flash column chromatography on silica gel (from 0 to 425% AcOEt-Hexanes).

Compound **2**: 40 mg, 0.053 mmol, 66%, [α]^20^_D_ = −9.13 (c = 0.5, CHCl_3_). *Rf* = 0.29 (Hexane/EtOAc, 5.5: 4.5), m.p: 149–150 °C. ^1^H-NMR (600 MHz, CDCl_3_): δ ppm 8.8–7.19 (m, 26H, H-arom ), 6.75 (d, 1H, *J*_2’,3’_ = 15.8 Hz, CH=CH-), 6.46–6.36 (m, 1H, CH=CH-), 5.98–5.88 (m, 2H, H-4, H-3), 5.27 (dd, 1H, *J*_1,2_ = J2, 3 = 2.7 Hz, H-2), 4.75 (dd, 1H, *J*_6a,6b_ = 12.0 Hz, J5, 6a = 6.9 Hz, H-6a), 4.50 (dd, 1H, *J*_6a,6b_ = 12.0 Hz, J_5,6b_ = 2.6 Hz, H-6b), 4.56–4.52 (m, 1H, H-1), 4.50–4.47 (m, 1H, H-5), 3.05–2.96 (m, 1H, H-1’a), 2.85–2.75 (m, 1H, H-1’b). ^13^CNMR (151 MHz, CDCl_3_): δ ppm, 166.2, 165.9, 165.6, 165.4 (4×CO), 150.1, 147.9, 135.89, 135.1, 133.5, 133.4, 132.9, 129.8, 129.6, 129.5, 128.59, 128.5, 128.2, 127.0, 121.3 (Carom, C-3’, C-2’), 74.2 (C-1), 71.2 (C-5), 71.1 (C-4), 69.6 (C-3), 68.0 (C-2), 62.9 (C-6), 33.1 (C-1’). ESI^+^-HRMS: [M + H]^+^ calcd for C_46_H_37_NO_9_ + H+: 748.2541; found, 748.2479.

Compound **3**: 44.6 mg, 0.064 mmol, 80 %. [α]^20^_D_ = −28.86 (c = 0.2, CHCl_3_). *R_f_* = 0.23 (Hexane/EtOAc, 1.8:1.2). ^1^H-NMR (600 MHz, CDCl_3_): δ ppm 8.44 (d, 1H, *^4^J*_H-H_ = 2.2 Hz, H-arom), 8.36 (dd, 1H, ^3^*J*_H-H_ = 4.7 Hz, ^4^*J*_H-H_ = 1.5 Hz, H-arom), 7.91–7.21 (m, 21H, H-arom), 7.11–6.93 (m, 1H, H-arom), 6.51 (d, 1H, *J*_2’,3’_ = 16 Hz, CH=CH-Py), 6.24 (m, 1H, *J*_2’,3’_ = 16.3 Hz, *J*_2,’H1’a_ = 13.4 Hz, *J*_2’,H1’b_ = 5.8 Hz CH=CH-Py), 6.02 (dd, 1H, *J*_3,4_ = *J*_4,5_ = 8.3 Hz, H-4), 5.81 (dd, 1H, *J*_2,3_ = 3.1 Hz, *J*_3,4_ = 8.6 Hz, H-3), 5.63 (dd, 1H, *J*_1,2_ = *J*_2_, _3_ = 3.4 Hz, H-2), 4.68 (dd, 1H, *J*_6a,6b_ = 12.0 Hz, *J*_5,6a_ = 6.7 Hz, H-6a), 4.52 (dd, 1H, *J*_6a,6b_ = 12. Hz, *J*_5,6b_ = 3.2 Hz, H-6b), 4.46–4.41 (m, 1H, H-1), 4.38–4.35 (m, 1H, H-5), 2.94–2.82 (m, 1H, H-1’a), 2.74–2.68 (m, 1H, H-1’b). ^13^CNMR (151 MHz, CDCl_3_): δ ppm, 166.2, 165.6, 165.6, 165.4 (4 × CO), 148.4, 148.2, 133.5, 133.4, 133.0, 132.4, 129.6, 128.9, 128.5, 126.7, 123.5 (Carom, C-3’, C-2’), 74.1 (C-1), 71.2 (C-5), 71.1 (C-2), 69.4 (C-4), 67.80 (C-3), 62.7 (C-6), 33.2 (C-1’). ESI^+^-HRMS: [M + H]^+^ calcd for C_42_H_35_NO_9_ + H+: 698.2385; found, 698.4013.

Compound **4**:115 mg, 0.152 mmol, 80% yield. [α]^20^_D_ = −9.92 (c = 0.5, CHCl_3_). *R_f_ =* 0.17 (Hexane/EtOAc, 2:1). ^1^H-NMR (300 MHz, CDCl_3_): δ, 8.21–7.00 (m, 24H, H-arom), 6.67 (d, 1H, *J*_2’,3’_ = 15.9 Hz, CH=CHPh), 6.39–6.2 (m, 1H, CH=CHPh), 6.00 (dd, 1H, *J*_3,4_ =*J*_4,5_ = 8.5 Hz, H-4), 5.91 (dd, 1H, *J*_2,3_ = 3.1 Hz, *J*_3,4_ = 8.9 Hz, H-3), 5.74 (dd, 1H *J*_1,2_ = *J*_2,3_ = 3.1 Hz, H-2), 4.66 (m, 2H, H-6a, H-6b), 4.74 (dd, 1H, *J*_6a, 6b_ = 12 Hz, *J*_5, 6a_ = 6.5 Hz, H-6a), 4.62 (dd, 1H, *J*_5, 6b_ = 12 Hz, *J*_6a, 6b_ = 2.8 Hz, H-6b), 4.59–4.36 (m, 2H, H-1, H-5), 3.09–2.92 (m, 1H, H-1’a), 2.85–2.78 (m, 1H, H-1’b), 2.37 (s,3H, Ac). ^13^C-NMR (75 MHz, CDCl_3_): δ 169.3, 166.2, 165.6, 165.4, 147.5 (5 × CO), 135.5, 133.4, 133.0, 129.8, 129.6, 129.5, 129.0, 128.4, 128.2, 126.7, 126.1, 122.5 (Carom, C-3’, C-2’), 74.4 (C-1), 71,4 (C-5), 71.0 (C-4), 69.7 (C-3), 67.8 (C-2), 62.8 (C-6), 33.23 (C-1’) and 21.1 (Ac). ESI^+^-HRMS: [M + H]^+^ calcd for C_45_ H_39_ O_11_, 755.2487; found, 755.2469.

Compound **5**: 49 mg, 0.064 mmol 80% yield. [α]^20^_D_ = −16.63 (c = 0.5, CHCl_3_)). *R_f_* = 0.22 (Hexane/Toluene/EtOAc, 71: 24: 5). ^1^H-NMR (300 MHz, CDCl_3_): δ, 8.14–7.29 (m, 22H, H-arom), 6.98 (dd, 1H, ^3^*J*_H-H_ = 9.0 Hz, ^4^*J*_H-H_ = 2.4 Hz, H-arom), 6.61 (d, 1H, *J*_2’,3’_ = 15.9 Hz, CH=CHPh), 6.32–6.17 (m, 1H, CH=CHPh), 5.95 (dd, 1H, *J*_3,4_ =*J*_4,5_ = 8.8 Hz, H-4), 5.90 (dd, 1H, *J*_2,3_ = 3.0 Hz, *J*_3,4_ = 8.8 Hz, H-3), 5.74 (dd, 1H *J*_1,2_ = *J*_2,3_ = 3.1 Hz, H-2), 4.72 (dd, 1H, *J*_6a,6b_ = 12 Hz, *J*_5,6a_ = 6.5 Hz, H-6a), 4.62 (dd, 1H, *J*_6a,6b_ = 12.1 Hz, *J*_5,6b_ = 2.8 Hz, H-6b), 4.56–3.36 (m, 2H, H-1, H-5), 3.05–2.87 (m, 1H, H-1’a), 2.83–2.79 (m, 1H, H-1’b), 2.32 (s, 3H, Ac). ^13^C-NMR (75 MHz, CDCl_3_): δ 169.3, 166.3, 165.6, 165.4, 147.5 (5 × CO), 135.4, 133.3, 133.0, 129.8, 129.5, 129.0, 128.4, 126.7, 126.1, 122.5 (Carom, C-3’, C-2’), 74.4 (C-1), 71,4 (C-5), 71.0 (C-4), 69.7 (C-3), 67.8 (C-2), 62.8 (C-6), 33.23 (C-1’) and 21.0 (Ac). ESI^+^-HRMS: [M + H]^+^ calcd for C_45_H_39_O_11_, 755.2487; found, 755.2453.

Compound **6**: 52 mg, 0.067 mmol, 90% yield. [α]^20^_D_ = −8.3 (c = 0.5, CHCl_3_). *R_f_* = 0.25 (Hexane/ EtOAc, 3:1). ^1^H-NMR (300 MHz, CDCl_3_): δ, 8.16.7.12 (m, 24H, H-arom), 6.63 (d, 1H, *J*_2’,3’_ = 15.8 Hz, CH=CHPh), 6.49–6.21 (m, 1H, CH=CHPh), 6.08–5.84 (m, 2H, H-4, H-3), 5.75 (dd, 1H *J*_1,2_ = *J*_2,3_ = 2.9 Hz, H-2), 5.11 (s, 2H, CH_2_Ph), 4.72 (dd,1H, *J*_6a,6b_ = 11.9 Hz, *J*_5,6a_ = 6.3 Hz, H-6a), 4.63 (dd, 1H, *J*_6a,6b_ = 11.9 Hz, *J*_5,6b_ = 2.6 Hz, H-6b), 4.57–4.39 (m, 2H, H-1, H-5), 3.01–2.93 (m, 1H, H-1’a), 2.84–2.77 (m, 1H, H-1’b), 2.84–2.77 (s, 3H, Ac). ^13^C-NMR (75 MHz, CDCl_3_): δ 166.2, 165.6, 165.6, 165.4, 137.6 (5 × CO), 134.9, 133.4, 133.0, 132.8, 129.8, 129.5, 128.9, 128.5, 126.3, 124.7 (Carom, C-3’, C-2’), 74.5 (C-1), 71,2 (C-5), 70.9 (C-4), 69.7 (C-3), 67.8 (C-2), 66.1 (CH_2_Ph), 63.0 (C-6), 29.6 (C-1’) and 21.0 (Ac). ESI^+^-HRMS: [M+NH_4_]^+^ calcd for C_46_H_44_NO_11_, 786.2909; found, 786.2919.

Compound **7**: 194 mg, 0.237, 73% yield. [α]^20^_D_ = −16.6 (c = 0.3, _CHCl3_). *Rf* = 0.18 (Hexane/EtOAc, 3:1). ^1^H-NMR (300 MHz, CDCl_3_): δ, 8.11–7.08 (m, 24H, H-arom), 6.65 (d, 1H, *J*_2’,3’_ = 15.9 Hz, CH=CHPh), 6.45–6.35 (m, 1H, CH=CHPh), 6.01–5.88 (m, 2H, H-4, H-3), 5.75 (dd, 1H, *J*_1,2_ = 3.9 Hz, *J*_2,3_ = 7.3 Hz, H-2), 4.78 (dd, 1H, *J*_6a,6b_ = 12.0 Hz, *J*_5,6a_ = 6.7 Hz, H-6a), 4.63 (dd, 1H, *J*_6a,6b_ = 12 Hz, *J*_5,6b_ = 2.8 Hz, H-6b), 4.58–4.42 (m, 2H, H-1, H-5), 3.93 (s, 3H, COOMe), 3.03–2.93 (m, 1H, H-1’a), 2.92.-2.71 (m, 1H, H-1’b). ^13^C-NMR (75 MHz, CDCl_3_): δ 166.8, 166.2, 165.7, 165.4, 141.4 (5 × CO), 133.4, 133.1, 132.4, 129.7, 129.6, 128.9, 128.5, 128.3, 127.1, 126.0 (Carom, C-3’, C-2’), 74.1 (C-1), 71.3 (C-5), 71.2 (C-4), 69.6 (C-3), 67.9 (C-2), 62.8 (C-6), 51.9 (COOMe), 33.1 (C-1’). ESI^+^-HRMS: [M + H]^+^ calcd for C_45_H_39_O_11_, 755.2487; found, 755.2493.

Compound **8**: 158 mg, 88% yield. [α]^20^D = −6.12 (c = 0.5, CHCl_3_). *R_f_* = 0.27 (Hexane/EtOAc, 3:1)^1^H-NMR (300 MHz, CDCl_3_): δ, 8.09–7.31 (m, 24H, H-arom), 6.65 (d, 1H, *J*_2’,3’_ = 15.9 Hz, CH=CHPh), 6.44 (m, 1H, CH=CHPh), 5.90 (m, 2H, H-4, H-3), 5.74 (dd, 1H *J*_1,2_ =3.7, Hz, *J*_2,3_ = 2.4 Hz, H-2), 4.86 (dd, 1H, *J*_6a,6b_ = 12 Hz, *J*_5,6a_ = 7.2 Hz, H-6a), 4.63–4.44 (m, 3H, H-6b, H-1, H-5), 3.12–2.98 (m, 1H, H-1’a), 2.76–2.85 (m, 1H, H-1’b). ^13^C-NMR (75 MHz, CDCl_3_): δ 166.2, 165.6, 165.4, 164.7 (4 × CO), 146.7, 143.3, (Carom-q),133.5, 133.1, 131.3, 129.8, 128.9, 128.3, 126.5, 123.6, (Carom, C-3’, C-2’), 73.4 (C-1), 71.4 (C-5), 71.0 (C-4), 69.4 (C-3), 68.2 (C-2), 62.6 (C-6), 33. 3 (C-1’). ESI^+^-HRMS: [M + Na]^+^ calcd for C_45_H_35_NO_11_, 764.2102; found, 764.2157.

Compound **10**: This compound was synthesized according to general procedure A. 56 mg, 0.072 mmol, 90% yield, [α]^20^_D_ = −14.16 (c = 0. 39, CHCl_3_). *Rf* = 0.27. (Hexane/EtOAc, 9:3). ^1^H-NMR (300 MHz, CDCl_3_): δ, 8.69 (d, 1H, ^3^*J*_H-H_ = 5.6 Hz), 8.10–7.23 (m, 23H, H-arom), 6.56 (d, 1H, *J*_2’,3’_ = 15.7 Hz, CH=CHPh), 6.33–6.12 (m, 1H, CH=CHPh), 6.01 (dd, 1H, *J*_3,4_ = 12 Hz, *J*_4,5_ = 9.0 Hz, H-4), 5.92 (dd, 1H, *J*_2,3_ = 3.1 Hz, *J*_3,4_ = 9.0 Hz, H-3), 5.15 (dd, 1H *J*_1,2_ = *J*_2,3_ = 3.1 Hz, H-2), 3.93–3.90 (m, 1H, H-1), 4.72 (dd, 1H, *J*_6a,6b_ = 12.1, Hz, *J*_5,6a_ = 6 Hz, H-6a), 4.58–4.41 (m, 2H, H-5, H-1), 3.09–2.98 (m, 1H, H-1’a), 2.93–2.76 (m, 1H, H-1’b). ^13^C-NMR (75 MHz, CDCl_3_): δ 166.2, 166.6, 165.6, 165.4, (4 × CO), 136.8, 133.3, 133.0, 132.8, 132.2, 129.7, 128.5, 128.9, 128.4, 127.5, 127.3, 127.1, 123.3 (Carom- C-3’,C-2’), (Carom), 127.5 (Carom, C-2’), 127.3, 127.1 (Carom), 74.5 (C-1), 71,3 (C-5), 70.9 (C-4), 69.7 (C-3), 67.8 (C-2), 62.9 (C-6), 32.9 (C-1’). ESI^+^-HRMS: [M + NH_4_]^+^: calcd for C_43_H_39_BrNO_9_, 794.1792; found, 794.1770

#### 3.2.2. General Procedure for De-O-Benzoylation

Compounds **11–18** and **20** were deprotected by treatment with 1M ammonia in methanol at room temperature for 36 h. Removal of solvent under vacuum afforded the crude residues which were purified by column chromatography (MeCN/MeOH, 9.5/0.5) and the compounds (11, 13, 15) were purified by semi-preparative HPLC (A: H_2_O + 0.1% trifluoroacetic acid, B: ACN + 0.1% trifluoroacetic acid, 5 mL/min).

Compound **11**: 15 mg, 0.045 mmol, 85% yield. [α]^20^_D_ = +41.3 (*c* = 0.1, CH_3_OH). ^1^H-NMR (600 MHz, CD_3_OD): δ ppm 8.87 (dd, 1H, ^3^*J*_H-H_ = 4.4 Hz, ^4^*J*_H-H_ = 1.6 Hz, H-arom), 8.33 (dd, 1H, ^3^*J*_H-H_ = 8.4 Hz, ^4^*J*_H-H_ = 1.6 Hz, H-arom), 7.97 (d, 2H, ^3^*J*_H-H_ = 1.1 Hz, H-arom), 7.84 (s, 1H, H-arom), 7.52 (dd, 1H, ^3^*J*_H-H_ = 8.3 Hz, ^4^*J*_H-H_ = 4.4 Hz, H-arom), 6.75 (d, 1H, *J*_2’,3’_ = 16.0 Hz, CH=CH-), 6.65–6.48 (m, 1H, CH=CH-), 4.07 (ddd, 1H, *J*_1,2_ = 2.7 Hz, *J*_1,H1’a_ = 8.7 Hz, *J*_1, H1’b_ = 6.0 Hz, H-1), 3.84–3.65 (m, 5H, H-4, H-3, H-2, H-6a H-6b), 3.64–3.55 (m, 1H, H-5), 2.84–2.69 (m, 1H, H-1’a), 2.68–2.51 (m, 1H, H-1’b). ^13^C-NMR (151 MHz, CD_3_OD): δ ppm: 149.2, 146.8, 136.9, (Carom-q), 136.4, 131.1, 128.4, 128.2, 127.6, 127.4, 125.2, 121.4, (Carom, C-3’, C-2’), 76.6 (C-1), 75,0 (C-5), 71.3 (C-4), 70.7 (C-3), 68.1 (C-2), 61.6 (C-6), 32.6 (C-1’). ESI^+^-HRMS: [M + H]^+^ calcd for C_18_H_21_NO_5_, 332.1492; found, 332.1515.

Compound **12**: 15 mg, 0.053 mmol, 84% yield. [α]^20^_D_ = +27.8 (*c* = 0.1, CH_3_OH). ^1^H-NMR (300 MHz, CD_3_OD): δ ppm 8.53 (s, 1H, H-arom), 8.38 (d, 1H, ^3^*J*_H-H_ = 3.8 Hz, H-arom), 7.94 (dd, 1H, ^3^*J*_H-H_ = 3.5 Hz, ^4^*J*_H-H_ = 1.8 Hz, H-arom), 7.42- (dd, 1H, ^3^*J*_H-H_ = 8.0 Hz, ^4^*J*_H-H_ = 4.9 Hz, H-arom), 6.60 (d, 1H, *J*_2’,3’_ = 16.0 Hz, CH=CH-Py), 6.50–6.41 (m, 1H, CH=CH-Py), 4.08 (ddd, 1H, *J*_1,2_ = 2.7 Hz, *J*_1_,_H1’a_ = 8.7 Hz, *J*_1_,_H1’b’_ = 6 Hz, H-1) 3.91–3.65 (m, 5H, H-4, H-3, H-2, H-6a, H-6b), 3.62–3.57 (m, 1H, H-5), 2.81–2.69 (m, 1H, H-1’a), 2.62–2.51 (m, 1H, H-1’b). ^13^C-NMR (75 MHz, CD_3_OD): δ ppm: 146.9, 146.6, 133.6, 129.3, 128.2, 124.2, 121.1 (Carom, C-3’, C-2’), 76.8 (C-1), 74.1 (C-5), 70.7 (C-4), 70.7 (C-3), 67.5 (C-2), 61.4 (C-6), 32.5 (C-1’). ESI^+^-HRMS: [M + H]^+^: calcd for C_14_H_19_NO_5_, 282.1336; found, 282.1331.

Compound **13**: 38 mg, 0.128 mmol, 85% yield. [α]^20^_D_ = +8.54 (c = 0.2, CH_3_OH). ^1^H-NMR (300 MHz, CD_3_OD): δ, 7.38 (dd, 1H, ^3^*J*_H-H_ = 8.3 Hz, ^4^*J*_H-H_ = 1.3 Hz, H-arom), 7.03 (td, 1H, ^3^*J*_H-H_ = 7.7 Hz, ^4^*J*_H-H_ = 1.6 Hz, H-arom), 6.76 (m, 3H, H-arom, CH=CHPh), 6.31–6.21 (m, 1H, CH=CHPh), 4.06–3.96 (m, 1H, H-1), 3.89–3.61 (m, 5H, H-4, H-3, H-6a, H-6b, H-2), 3.56 (ddd, 1H, *J*_4,5_ = 8.5 Hz, *J*_5,6a_ = 5.4 Hz, *J*_5,6b_ = 2.9 Hz, H-5), 3.01 (s, 1H, PhOH), 2.66 (ddd, 1H, 1H, *J*_H1’a, H1’b_ = 14.7 Hz, *J*_1,H1’a_ = 1.4 Hz, *J*_1’,2’_ = 8.1 Hz, H-1’a), 2.54 (ddd, 1H, 1H, *J*_H1’a,H1’b_ = 14.4 Hz, *J*_1,H1’b_ = 3.9 Hz, *J*_H1’b,2’_ = 10.6 Hz, H-1’b). ^13^C-NMR (75 MHz, CD_3_OD): δ 154.1 (C-PhOH), 127.5 (C-3’), 127.3, 126.2 (Carom-q), 122.0 (C-2’), 124.4, 118.9, 115.0 (Carom), 77.5 (C-1), 74,8 (C-5), 71.4 (C-4), 70.6 (C-3), 67.9 (C-2), 61.6 (C-6), 33.09 (C-1’). ESI^+^-HRMS: [M+Na]^+^: calcd for C_15_H_20_NaO_6_, 319.1158; found, 319.1175.

Compound **14**: 16 mg, 0.054 mmol, 85% yield. [α]^20^_D_ = +31 (c = 0.1, CH_3_OH). ^1^H-NMR (300 MHz, CD_3_OD): δ, 7.23 (d, 2H, ^3^*J*_H-H_ = 8.6 Hz, H-arom), 6.72 (d, 2H, ^3^*J*_H-H_ = 8.6 Hz, H-arom), 6.42 (d, 1H, *J*_2’,3’_ = 15.7 Hz, CH=CHPh), 6.14–6.04 (m, 1H, CH=CHPh),4.10–3.91 (m, 1H, H-1), 3.87–3.61 (m, 5H, H-4, H-3, H-6a, H-6b, H-2), 3.62–3.44 (m, 1H, H-5), 2.66–2.57 (m, 1H, H-1’a), 2.53–2.43 (m, 1H, H-1’b). ^13^C-NMR (75 MHz, CD_3_OD): δ 153.6 (C-PhOH), 131.7 (C-3’), 129.1, 126.9 (Carom-q), 122.6 (C-2’), 115.1 (Carom), 77.5 (C-1), 74,8 (C-5), 71.4 (C-4), 70.6 (C-3), 67.9 (C-2), 61.6 (C-6), 33.09 (C-1’). ESI^+^-HRMS: [M+Na]^+^: calcd for C_15_H_20_NaO_6_, 319.1152; found, 319.1165.

Compound **15**: 16.2 mg, 0.052 mmol, 80% yield. [α]^20^_D_ = +20 (c = 0.1, CH_3_OH). ^1^H-NMR (300 MHz, CD_3_OD): δ, 7.38 (d, 2H, ^3^*J*_H-H_ = 8.2 Hz, H-arom), 7.29 (d, 2H, ^3^*J*_H-H_ = 8.2 Hz, H-arom), 6.52 (d, 1H, *J*_2’,3’_ = 15.9 Hz, CH=CHPh), 6.39–6.24 (m, 1H, CH=CHPh),4.58 (s, 2H, CH_2_Ph), 4.07–3.94 (m, 1H, H-1), 3.86–3.61 (m, 5H, H-4, H-3, H-6a, H-6b, H-2), 3.56 (ddd, 1H, *J*_4,5_ = 8.5 Hz, *J*_5,6a_ = 5.5 Hz, *J*_5,6b_ = 3.0 Hz, H-5), 2.72–2.61 (m, 1H, H-1’a), 2.57–2.61 (m, 1H, H-1’b). ^13^C-NMR (75 MHz, CD_3_OD): δ 140.2,136.6 (Carom-q), 131.8, (C-3’), 126.8, 125.7 (Carom), 125.6, (C-2’), 77.2 (C-1), 74,9 (C-5), 71.2 (C-4), 70.7 (C-3), 68.0 (C-2), 63.5 (CH_2_Ph), 61.6 (C-6), 32.4 (C-1’). ESI^+^-HRMS: [M+NH_4_]^+^: calcd for C_16_H_26_NO_6_, 328.1416; found, 328.1771.

Compound **16**: 16 mg, 0.047 mmol, 85% yield. [α]^20^_D_ = +42.62 (c = 0.1, CH_3_OH). ^1^H-NMR (300 MHz, CD_3_OD): δ, 7.95 (d, 2H, ^3^*J*_H-H_ = 8.4 Hz, H-arom), 7.51 (d, 2H, ^3^*J*_H-H_ = 8.4 Hz, H-arom), 6.64–6.44 (m, 2H, CH=CHPh, CH=CHPh), 4.04 (ddd, 1H, *J*_1,2_ = 6.0 Hz, *J*_1,1’a_ = 8.7 Hz, *J*_1,1’b_ = 2.7 Hz, H-1), 3.83 (dd, 1H, *J*_3,4_ =6.1 Hz, *J*_4,5_= 3.1 Hz, H-4,), 3.81–3.64 (m, 4H, H-3, H-6a, H-6b, H-2), 3.55 (ddd, 1H, *J*_4,5_ = 9.9 Hz, *J*_5,6a_ = 6.5 Hz, *J*_5,6b_ = 3.8 Hz, H-5), 2.76–2.66 (m, 1H, H-1’a), 2.64–2.47 (m, 1H, Hb). ^13^C-NMR (75 MHz, CD_3_OD): δ 167.0 (COOMe), 142.3 (Carom-q), 131.0 (C-3’), 129.4 (Carom-q), 128.2, 125.9 (C-2’), 77.1 (C-1), 75,2 (C-5), 70.9 (C-4), 70.7 (C-3), 67.8 (C-2), 61.5 (C-6), 51.4 (COOMe), 32.6 (C-1’). ESI^+^-HRMS: [M+Na]^+^: calcd for C_17_H_22_NaO_7_, 361.1258; found, 361.1253.

Compound **17**: 16 mg, 0.049 mmol, 85% yield. [α]^20^_D_ = +40.6 (c = 0.1, CH_3_OH). ^1^H-NMR (300 MHz, CD_3_OD): δ, 8.19 (d, 2H, ^3^*J*_H-H_ = 5 Hz, H-arom), 7.64 (d, 2H, ^3^*J*_H-H_ = 4.9 Hz, H-arom), 6.95–6.45 (m, 2H, CH=CHPh, CH=CHPh), 4.26–3.89 (m, 1H, H-1), 3.84–3.54 (m, 6H, H-4, H-3, H-6a, H-6b, H-2, H-5), 3.10–2.64 (m, 1H, H-1’a), 2.71–2.37 (m, 1H, H-1’b). ^13^C-NMR (75 MHz, CD_3_OD): δ 146.4, 144.3 (Carom-q), 132.1, 129.9, 126.2, 123.7 (Carom, C-3’, C-2’), 76.5 (C-1), 74.9 (C-5), 71.1 (C-4), 76.5 (C-3), 67.9 (C-2), 61.2 (C-6), 32.4 (C-1’). ESI^+^-HRMS: [M+NH_4_]^+^: calcd for C_15_H_23_N_2_O_7_, 343.15; found, 343.1512.

#### 3.2.3. Synthesis of Compounds **19** and **20** by Suzuki Reaction

To a solution of compound **10** (40 mg, 0.051 mmol) in degassed (N_2_) dioxane-water (5:1) were added 4-piridinylboronicacid (10 mg, 2 equiv.), 15% Pd_2_Cl_2_ (PPh_3_)_2_ ferrocene, and cesium carbonate (48 mg, 3 equiv.). The reaction mixture was heated at 80 ^0^C. The reaction was followed by TLC. The solution was evaporated under reduced pressure and the residue was purified by flash column chromatography on silica gel (6:4 Hexane/EtOAc) to afford compound **19** as colorless oil. 22 mg, 0.028 mmol, 55%, [α]^20^_D_ = −10.7 (c = 0. 4, CHCl_3_). *Rf* = 0.17 (Hexane/EtOAc, 6:4). ^1^H-NMR (300 MHz, CDCl_3_): δ, 8.69 (d, 1H, H-arom), 8.06.7.23 (m, 27H, H-arom), 6.56 (d, 1H, *J*_2’,3’_ = 15.6 Hz, CH=CHPh), 6.28 (ddd, 1H, *J*_1’a,2’_ = *J*_1’b_, 2’=6.9 Hz, *J*_2’,3’_ = 15.6 Hz, CH=CHPh), 6.01 (dd, 1H, *J*_3,4_ = *J*_4,5_ = 8.6 Hz, H-4), 5.92 (dd, 1H, *J*_2,3_ = 3.3 Hz, *J*_3,4_ = 9.0 Hz, H-3), 5.72 (dd, 1H *J*_1,2_ = *J*_2,3_ = 3.2 Hz, H-2), 4.71 (dd, 1H, *J*_6a,6b_ = 12.1 Hz, *J*_5,6a_ = 5.9 Hz, H-6a), 4.62 (dd, 1H, *J*_6a,6b_ = 12.2 Hz, *J*_5,6b_ = 2.9 Hz, H-6a), 4.52–4.44 (m, 1H, H-1), 4.41–4.34 (m, 1H, H-5), 2.94–2.68 (m, 2H, H-1’a, H-1’b). ^13^C-NMR (75 MHz, CDCl_3_): δ 166.2, 165.6, 165.5, 165.4, (4 × CO), 148.9 (Carom-N), 137.3, 135.0, 133.5, 133.4, 133.3, 133.0, 131.3, 129.8, 129.7, 128.9, 128.5, 127.7, 126.5, 125.2 (Carom, C-3’, C-2’), 74.5 (C-1), 70,4 (C-5), 69.1 (C-4), 67.8 (C-3), 66.0 (C-2), 62.8 (C-6), 33.3 (C-1’). ESI^+^-HRMS: [M + H]^+^: calcd for C_48_H_40_NO_9_, 774.26; found, 774.2690.

Compound **19** was deprotected according to general procedure above to afford compound **20**: 7 mg, 0.02 mmol, 80%, [α]^20^_D_ = +17.38 (*c* = 0.13, CH_3_OH). ^1^H-NMR (300 MHz, CD_3_OD): δ 8.61 (s, 2H, Harom), 7.69 (d, 1H, ^3^*J*_H-H_ = 7.9 Hz, H-arom), 7.49–7.29 (m, 5H, ^3^*J*_H-H_ = 7.9 Hz, H-arom), 6.43 (d, 1H, *J*_2’,3’_ = 15.7 Hz, CH=CHPh), 6.33–6.14 (m, 1H, CH=CHPh), 4.04–3.91 (m, 1H, H-1), 3.83–3.62 (m, 5H, H-4, H-3, H-2, H-6a,H-6b) 3.5–3.47 (m, 1H, H-5), 2.69–2.51 (m, 1H, H-1’a), 2.50–2.38 (m, 1H, H-1’b). ^13^C-NMR (151 MHz, CD_3_OD): δ, 148.6, 136.9, 135.4, 129.9 (Carom), 129.1 (C-2’), 128.7, 128.6 (C-3’), 128.0, 127.2, 126.3, 125.1 (Carom), 76.5 (C-1), 74,8 (C-5), 71.1 (C-4), 70.6 (C-3), 68.9 (C-2), 61.7 (C-6), 32.5 (C-1’). ESI^+^-HRMS: [M + H]^+^: calcd for C_20_H_24_NO_5_, 357.16; found, 357.1666.

### 3.3. Bioactivity Assay

#### 3.3.1. Expression and Purification of FimH

The FimH lectin (Phe1-Thr158) was produced and purified as described previously [46] by expression from the pET-24a vector in *E. coli* C43(DE3) [45] in MinA medium complemented with the 20 amino acids, the vitamins biotin and thiamin, glucose and MgCl_2_. Soluble FimH lectin secreted in the periplasm was extracted by applying 30% sucrose and 2.5 mM EDTA, in 20 mM HEPES at pH 7.4, onto the washed bacterial pellet followed by a 30-fold dilution in the same buffer to cause the desired osmotic shock. A 30’ centrifugation at 13,000× *g* was carried out to eliminate the cellular debris present in the pellet and isolate the soluble proteins in the supernatant. The supernatant was acidified using HCl to pH = 3.9 and centrifuged again before cation exchange chromatography onto an HiTrap Sulfopropyl Fast Flow column (SPFF, Cytiva). The SPFF column was washed in 20 mM formic acid (pH = 3.9) and eluted using a salt gradient. The fractions containing FimH lectin eluted between 150–250 mM NaCl and were immediately neutralized upon elution by adding a drop of 1 M HEPES at pH 7.4. Finally, pure protein fractions were pooled and dialyzed against 20 mM HEPES at pH 7.4 containing 150 mM NaCl.

#### 3.3.2. Co-Crystallization of Antagonist **11** with FimH

Purified FimH lectin was concentrated to 17.19 mg·mL^−1^ and 1 mM of ligand **11** was added for co-crystallization at 20 °C using the sitting-drop vapor-diffusion method. A single crystal was obtained in a condition from the JCSG crystallization screen (Molecular Dimensions), containing 3.0 M NaCl and 0.1 M BIS-TRIS at pH = 5.5 (Appendix A). Cryoprotection prior to flash-freezing in liquid nitrogen was performed by dragging the crystal through a drop containing 3.5 M NaCl, 50 mM BIS-TRIS at pH 5.5 and 30% glycerol. The crystal diffracted to 3 Å resolution at the PX1 beamline of the French Soleil synchrotron. Molecular replacement using PDB entry 2VCO (same as above) [46] of its oligomannose-3 ligands, led to the placement of four FimH lectin protomers in the unit cell of the hexagonal crystal. Iterative rounds of refinements were performed using PHENIX and the model was adjusted manually using Coot [56]. The crystal structure was run through PDB_REDO [57], for further optimization, and validated using Molprobity [58] and Staraniso [59].

#### 3.3.3. Affinity Evaluation of Mannosides through FimH LEctPROFILE Kit

FimH LEctPROFILE kit assays from GLYcoDiag (Orléans, France) were performed according to GlycoDiag’s protocol already described [60,61,62]. Briefly, the interaction profiles of each compound were determined through a competitive inhibition assay based on the inhibition by the compounds of the interaction between FimH lectin coated onto the microplate surface and a biotinylated neoglycoprotein NeoM (Man-BSA) as a tracer. A mix of biotinylated Man-BSA (fixed concentration) and the corresponding compounds (range of concentrations) prepared in PBS supplemented with 1 mM CaCl_2_ and 0.5 mM MgCl_2_ was deposited in each well (50 µL each) in triplicate and incubated for two hours at room temperature. After washing with PBS buffer, the conjugate streptavidin-DTAF (dichlorotriazinylamino fluorescein) was added (50 µL) and incubated 30 min more. The plate was washed again with PBS. Finally, 100 µL of PBS was added for the readout of fluorescent plate performed with a fluorescence reader (Pherastar microplate reader, BMG labtech, λex = 485 nm, λem = 530 nm). The signal intensity is inversely correlated with the capacity of the compound to be recognized by the lectin and expressed as inhibition percentage with comparison with the corresponding tracer alone. Data analysis was performed with GraphPadPrism software (version 5.03 for windows, San Diego, CA, USA). 50% inhibitory concentration (IC_50_) was determined according to a standard dose-response/inhibition fitting model with the following equation: y = 100 / (1 + [inhibitor]/IC_50_) and expressed in nanomolar units.

#### 3.3.4. Bacterial Strains and Cell Line

The well characterized UPEC strain CFT073 (ATCC 700928) was used as the uropathotype in this study. Strain CFT073 was grown at 37 °C in LB or seeded onto MacConkey agar plates. The presence of the *fimH* gene was confirmed by PCR using the primers fimH-F 5′-TGCAGAACGGATAAGCCGTGG-3′ and fimH-R 5′-GCAGTCACCTGCCCTCCGGTA-3′ and *E. coli fimH*-proficient and -deficient strains served as positive and negative controls (E1P and I2P strains), respectively [5,63,64]. The human bladder epithelial cell line 5637, (ATCC HTB-9) (ATCC-LGC, Milan, Italy) was routinely cultured in T25 flasks at 37 °C in a humidified atmosphere with 5% CO_2_ using Roswell Park Memorial Institute (RPMI) 1640 medium supplemented with 10% FBS (both Gibco, Milan, Italy).

#### 3.3.5. Effect of Mannosides on Bacterial Growth and Metabolism

Mannosides dissolved in dimethyl sulfoxide DMSO or water at a final concentration of 5 mM were prepared. To test the mannosides’ toxicity, strain CFT073 was grown on LB medium supplemented with each molecule reported in Table 1, at a final concentration of 50, 100 and 500 μM. Natural D-mannose (D-Man) at equal concentration to the mannosides was used as control. LB supplemented with DMSO was included as growth control. Bacterial cultures were then incubated in a 96-well plate at 37 °C over a period of 10 h with a 30 min temperature equilibration period before data acquisition started. Readings of culture turbidity (OD_600_) were determined using a plate reader (POLARstar Omega BMG Labtech plate reader, Germany). To evaluate whether mannosides could be used as carbon sources, an initial inoculum of strain CFT073 of ~1.5 × 10^7^ CFU/mL was incubated in PBS supplemented with each molecule at a final concentration of 500 μM for 24 h at 37 °C. Following incubation, the growth of strain CFT073 was determined by colony forming unit (CFU/mL) counting by spot-plating serial dilutions.

#### 3.3.6. Cell Viability and Toxicity Assay

To evaluate whether synthesized mannosides can affect eukaryotic cells viability, HTB-9 cells were seeded onto 24-well plates at 5 × 10^5^ cells per mL and incubated in RPMI supplemented with 10% FBS in the presence of each mannoside molecule at a final concentration of 50, 100 and 500 μM at 37 °C in a 5% CO_2_ atmosphere for 24 h. At this point, cell viability was determined by the MTT assay. The medium was replaced by fresh RPMI supplemented with 10% FBS and 1 mg/mL MTT, and the cells were further incubated for 1 h. Viable cells, with active metabolism able to metabolize yellow tetrazole (MTT) into purple formazan crystals, were quantified by measuring the absorbance at 570 nm of formazan crystals formed and solubilized with isopropanol.

#### 3.3.7. Antibiotic-Mannoside Interference Assay

To assess any interference with antibiotic activities, strain CFT073 (approximately 10^5^–10^6^ CFU/well) was inoculated into a 96-well microplate supplemented with LB containing ampicillin (AMP 30 μg/mL), streptomycin (SM 50 µg/mL) and gentamycin (GM 50 µg/mL) with or without the addition of each mannoside antagonist (500 μM concentration). LB supplemented with DMSO was included as growth control in this experiment. The microplate was incubated at 37 °C and bacterial growth kinetics were monitored by measuring the OD_600_ over a period of 16 h.

#### 3.3.8. Bacterial Adhesion Assay

HTB-9 cells were routinely seeded in cell culture plates and maintained 2–4 days at 37 °C in a humidified atmosphere containing 5% CO_2_. For the adhesion assay, cells were seeded in 35 mm tissue culture plates at a density of 1 × 10^5^ cells/well and incubated at 37 °C for 48 h to reach confluency. CFT073 was grown in LB under mild shaking conditions overnight and resuspended in phosphate buffer (PBS) to an inoculum of ~10^6^ CFU/mL (normalized according to OD_600_). Each mannoside molecule was added to CFT073 inoculant at the final concentrations of 100, 500 μM and 1 mM and incubated for 3 h in static conditions. One ml of these bacterial/mannosides mixtures was used to infect HTB-9 cell monolayers at a multiplicity of infection (MOI) of 10; monolayers were centrifuged (10’ at 2000× *g*) and incubated at 37 °C with 5% CO_2_ for 2.5 h. The CFT073 strain incubated without any mannoside molecules was used as control. Monolayers were extensively washed (seven times) with PBS and lysed with 0.1% Triton X-100 in PBS. Cell lysates were serially diluted and spot-plated onto LB agar plates for CFU/mL counting. Parallel infected cells were Giemsa stained for qualitative assessment of bacterial adhesion, as previously described [5,65]. Images were recorded with a Leica DM5000B microscope equipped with DFX340/DFX300 camera and processed using the Leica Application Suite 2.7.0.R1 software (Leica).

#### 3.3.9. Molecular Dynamics Simulations

The complex of compound **20** and the FimH lectin domain (a.a. 1-158) with the best score using induced fit was used as the starting configuration for the molecular dynamics (MD) simulations. The complex was solvated and the structural waters were added using the same structural information as in the docking (PDB code: 4AUY) [66] and the ionic concentration was set to 0.15 M NaCl. In accordance with propKa [67] the standard protonation state at pH 7 was used for all protonatable groups of FimH. The generated molecular system comprised about 45,000 atoms including around 15,000 water molecules. The CHARMM36 force field with CMAP corrections was used to describe protein, water, and ion atoms [68,69,70]. Missing force field parameters for compound **20** were initially generated with CGenFF [68] with standard parameters and afterwards adapted. The integrity of the compound was verified in a 50-ns long MD simulation of the compound alone in water using the adapted force field.

Two independent simulations of the so generated system were performed. In each of them a three-step equilibration was applied: first, a 2.5 ns long equilibration of the water and ions molecules, second a 2.5 ns long equilibration in which only the protein backbone was fixed, and third unrestrained simulations was carried out for 2.5 ns. This was followed by a 30-ns long production run. 

All MD calculations were performed in the isothermal-isobaric ensemble at 300 K with the program NAMD2.9 [48]. Long-range electrostatic interactions were calculated using the particle-mesh Ewald method [49]. A smoothing function was applied to truncate short-range electrostatic interactions. The Verlet-I/r-RESPA multiple time-step propagator [51] was used to integrate the equation of motions using a time step of 2 and 4 fs for short- and long-range forces, respectively. All bonds involving hydrogen atoms were constrained using the Rattle algorithm [50]. 

## 4. Conclusions

In this study, the design, synthesis, and function of a small library of *C*-mannose inhibitors containing heteroaryl moieties were reported. Their relative binding affinity was measured using a competitive inhibition assay against the binding of FimH with mannosylated BSA conjugate. Among them, the best results were obtained with compounds **20** and **11**, respectively. Although compound **20** had a higher inhibitory potency (IC_50_ 0.82 ± 0.4) against isolated FimH lectin binding domain, ligand **11** (IC_50_ 3.17 ± 2.3) showed better potency in inhibiting bacterial adhesion to bladder HTB-9 cell monolayers without adverse side effects. The crystal structure of the FimH lectin-binding domain co-crystallized with inhibitor **11** was obtained at a resolution of 3 Å. It also confirmed the expected *^4^C_1_* chair conformation of antagonist **11** bound into the active site of the tyrosine gate. Interestingly, the inter-molecular π-π stacking of two quinolines residues of **11** triggered the interlacing of two FimH lectins, providing a “bidentate’’ complex. Furthermore, in the mannose-binding site between FimH and ligand **11**, Tyr48 was shown to be p-stacked in parallel to the quinoline moiety, while its Tyr137 appeared to form T-aromatic stacking. On the other hand, molecular dynamics (MD) simulations were done for the best antagonist **20**, which unfortunately failed to provide co-crystals with FimH. The results indicated that the lowest potential energy was obtained with the FimH in its *half-open* conformation. The docking of compound **20** to this conformer helped raising the hypothesis that its high affinity may originate from a p-stacking of the first phenyl ring with Tyr48, as well as from the interaction of the *ortho*-pyridyl moiety with Tyr137 through a potential hydrogen bond.

In addition, the synthetic *C*-linked mannopyranoside inhibitors discussed herein neither affected bacterial growth or cell viability, nor interfered with antibiotic activity. The latter aspect is particularly important because antibiotics still represent the standard treatment for UTIs. However, literature data evidenced the increase of the number of cleared infections when antibiotics were administrated in combination with D-mannose [71]. Moreover, the preventive use of D-mannose showed a reduced number of UTIs in patients suffering from rUTIs. Hence, the reported mannoside derivatives, and in particular molecules **11** and **20**, represent good candidates to be analyzed in clinical trials to definitively accelerate the inclusion of mannoside-based FimH inhibitors in the clinical guidelines for the treatment of UTIs. 

## Data Availability

Not applicable.

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
