# Peer review of "Insightful Improvement in the Design of Potent Uropathogenic E. coli FimH Antagonists"

_pharmaceutics, 2023, doi:10.3390/pharmaceutics15020527_

Round 1

Reviewer 1 Report

Please modify the title so as to remove the word "Further". It gives impression as if there are 2 or more manuscripts by same authors being published simultaneously in the same journal, although the work being referred here in from 2017.

Figure 5 and Figure 7 donot co-relate. As per Figure 7, natural D Mannose supports the growth of strain many folds, while as per figure 5, D-Mannose supplemented cells seem to be late starters. For Figure 5, was the N-mannose control run on the same 96 well plate as rest of the samples plotted in this figure?  Or was the N-Mannose growth curve done separately and then plotted together with the samples to get figure 5? If done along with rest of the samples shown, the original plot obtained by the plate reader can be submitted as supplementary file to support. If this is not the case, all samples and control must be run together in same 96 well plate and then plotted.

Author Response

Please modify the title so as to remove the word "Further". It gives impression as if there are 2 or more manuscripts by same authors being published simultaneously in the same journal, although the work being referred here in from 2017.

Reply:

Although the term ‘’Further’’ does not necessarily mean what the reviewer implies, we changed it for ‘’Insightful development in the design of....’’

Comment 2:

Figure 5 and Figure 7 do not co-relate. As per Figure 7, natural D Mannose supports the growth of strain many folds, while as per figure 5, D-Mannose supplemented cells seem to be late starters. For Figure 5, was the N-mannose control run on the same 96 well plate as rest of the samples plotted in this figure?  Or was the N-Mannose growth curve done separately and then plotted together with the samples to get figure 5? If done along with rest of the samples shown, the original plot obtained by the plate reader can be submitted as supplementary file to support. If this is not the case, all samples and control must be run together in same 96 well plate and then plotted.

Reply:

The authors thank this reviewer’s comment that allows us to clarify our experimental procedures and results. The main purpose of experiments presented in figure 5 was to assess whether mannosides could exert a toxic activity towards bacterial cells. To this aim, bacteria were cultivated into a nutrient rich medium (Luria-Bertani broth, LB) in the absence and in the presence of the different mannosides. The LB medium alone and LB supplemented with the natural D-mannose were used as positive controls of bacterial growth. No statistically significant differences were observed in all condition tested demonstrating that mannosides do not interfere with bacterial replication. Hence, mannosides do not show any toxic activity on bacterial cells. Bacterial cultures were performed in multiwell plates including both the controls and the samples in each experimental replicate. Differently, the main purpose of experiments in figure 7 was to evaluate if mannosides could be used as carbon sources and could sustain bacterial replication as already observed for the natural D-mannose. To this aim bacteria were cultured in LB for 12 hours and then diluted to 1.5 x 107 CFU/mL in PBS buffer and incubated in the absence of any other carbon sources except of mannosides for 24 hours at 37°C. PBS alone (non-treated) was used as negative control of bacterial growth vice versa, PBS supplemented with two different D-mannose concentrations (500 µm and 83 mM) was used as positive control. After the incubation period, bacteria were serially diluted and plated onto LB agar plates for CFU/mL counting. Results show that only D-mannose is metabolized and sustains bacterial replication as determined by the higher CFU/mL counting compared to non-treated as well as to mannoside-treated bacteria. Hence, mannosides do not allow for bacterial replication and this represents an important advantage for their clinical use.

The text of section 2.5 was thus modified accordingly.

Reviewer 2 Report

Dear editor and authors Concerning the MS; Further insights in the design of potent Uropathogenic E. coli Submitted to pharmaceutics-2182684 by Leila Mousavifar , Meysam Sarshar , Clarisse Bridot , Daniela Scribano , Cecilia Ambrosi , Anna Teresa Palamara , Gérard Vergoten , Benoît Roubinet , Ludovic Landemarre , Julie Bouckaert * , Rene Roy The MS discussed the development of  Selective antagonists for the Uropathogenic Escherichia coli (UPEC) type-1 fimbrial adhesin (FimH) The MS is well written, with clear approved data.  FimH antagonist binds with the active site of the tyrosine gate. without affecting bacterial growth or cell viability nor interfered with antibiotic activity Minor points; More details concerning UPEC need to be added to the introduction. More data about previous work of UPEC antagonists need to be added. All the figure the line need to be more size, and uniform.  

Author Response

Comment 1:

Minor points; More details concerning UPEC need to be added to the introduction.

Reply:

We added several sentences marked in yellow while the referencing was considered satisfactory.

More data about previous work of UPEC antagonists need to be added.

 Reply:

Given that this is a research paper and not a review, we considered that the set of previous work by other researchers in the field using cited references 14-27, in completion to our own review paper (Ref 4 in Acc. Chem. Res.)  provides a proper overall view of this topic. Major actors (Hultgren, Janetka, Lindhorst, Gouin, Ernst) in this field were, in our opinion adequately cited.

All the figure the line need to be more size, and uniform.

Reply:

We agreed with this valuable suggestion by the reviewer and we changed all the requested Figures accordingly. 

Reviewer 3 Report

Dear authors, 

Congratulations to your excellent work.

Author Response

Congratulations to your excellent work.

Reply:

We are thankful to this reviewer for recognizing the nice coverage and execution of this work.

Note to the editor: Modern grammatical and typographic spell checker was used throughout. Notice that we have also made extensive english style revisions with the aid of a native english individual. Several changes are marked in yellow